

# Enhanced recurrent attention-deep Q learning with optimal node constrains and effective penalty based model for data transmission scheduling on wireless sensor networks

D.R. Anita Sofia Liz[1] and Yesubai Rubavathi C[2]

[1] CSE, New Prince Shri Bhavani College of Engineering and Technology, Chennai, Tamil Nadu, India
[2] CSE, Saveetha Engineering College, Chennai, Tamil Nadu, India

## ABSTRACT

Effective scheduling of data transmission is critical to maximizing network performance and resource usage in the context of wireless sensor networks (WSNs). In order to improve the effectiveness of data transmission scheduling in wireless sensor networks (WSNs), this paper proposes a unique method called Recurrent Attention-Deep Q Learning with Optimal Node Constraints and Effective Penalty based WSN Scheduling (RA-DQL-ONC&EP). This technique performs dynamic scheduling of data transmission tasks considering energy consumption and network interference by combining a penalty-based model, optimal node limitations, and recurrent attention techniques. Simulation results show that the proposed approach performs remarkably well. With a 91.21% success rate, it also guarantees dependable data transport throughout the network. Additionally, the delay rate is reduced to 1.99%, demonstrating effective data transfer with low latency. It is an effective model, yet it uses 70% less energy than other models since it is energy-efficient. The algorithm's performance is further demonstrated by throughput analysis, which shows a 72% throughput over 1,000 time steps. Based on enhanced reliability, efficiency, and energy conservation in network operations, our results highlight the potential of RA-DQL-ONC&EP as a promising approach for improving data transmission scheduling in WSNs. This optimized scheduling enhances network reliability, ensuring timely and accurate data delivery, which can support various applications such as environmental monitoring, healthcare systems, and smart city infrastructure, ultimately fostering societal well-being and progress. Additionally, the algorithm's efficiency contributes to cost savings and resource conservation, making it a socially responsible choice for managing wireless sensor networks.

Corresponding author
D.R. Anita Sofia Liz,
anitasofializphd@gmail.com

# INTRODUCTION

Wireless sensor networks (WSNs) have emerged as a transformative technology across various fields, enabling the seamless integration of sensors for real-time data collection and transmission. Effective data transfer scheduling is critical in WSNs to ensure timely and reliable communication between sensor nodes. In this context, Deep Q-Learning (DQL) algorithms have garnered significant attention for enhancing real-time data transmission scheduling within WSNs. DQL, a subclass of reinforcement learning, employs neural networks to learn optimal policies through trial-and-error interactions with the environment. Researchers anticipate that incorporating DQL into WSNs will significantly enhance the network's ability to dynamically adapt and optimize data transmission schedules. This innovative approach addresses critical challenges such as limited energy resources, fluctuating network conditions, and dynamic data traffic patterns. The application of DQL in WSNs not only optimizes resource allocation but also improves overall network performance, leading to reduced latency, enhanced reliability, and extended network lifetime. The convergence of WSNs and DQL holds substantial promise for a range of applications, including smart cities, environmental monitoring, healthcare systems, and other domains where real-time data transmission is vital for informed decision-making and operational efficiency. As research in this area progresses, the synergy between WSNs and DQL continues to unlock new possibilities, offering scalable solutions to the evolving demands of modern information systems.

A multi-objective intelligent clustered routing schema (*Ghamry & Shukry, 2024*) has been proposed for IoT-enabled WSNs, employing deep reinforcement learning. The integration of deep reinforcement learning addresses key challenges in WSN by enhancing energy efficiency, packet delivery, and network longevity. The operational model partitions the network into unequal clusters based on data load, thereby preventing premature network failure. This uneven clustering scheme is structured to balance inter- and intra-cluster energy consumption among cluster heads. Simulation results show that the proposed system outperforms existing schemes in terms of energy efficiency, packet delivery ratio, end-to-end latency, number of alive nodes, and network lifetime. However, the specifics of dataset used along with the particular limitations and advantages of the proposed scheme, are not explicitly detailed in the provided material. In addition, a deep learning-based heterogeneous data clustering technique (*Sudhakar & Anne, 2024*) optimizes data processing for edge-enabled IoT devices in agriculture. The integration of deep reinforcement learning into an edge-enabled wireless sensor network enables multi-objective optimization, effectively addressing task scheduling and resource allocation challenges in dynamic agricultural environments. The Deep Q-Networks (DQNs) framework, which employs a deep neural network to approximate the $Q$-value function, demonstrates superior performance compared to existing methods when handling complex issues in high-dimensional state spaces. In simulated smart farming applications, the framework surpasses other approaches in terms of energy consumption, latency, resource utilization, and task completion time. Nevertheless, the material provided does not include

detailed information regarding the dataset used or any potential drawbacks of the propose method.

The crucial problem of information infrastructure overload during catastrophes is addressed by proposing a multi-unmanned aerial vehicle (UAV) scheduling method for data collecting. This study (*Wan et al., 2024*) presents an attention-based deep reinforcement learning (DRL) algorithm designed to detect the time-varying value of disaster data. The application of a specialized team navigation problem model further enhances computational efficiency. The research highlights the essential role of UAVs as mobile relays and emphasizes the advantages of attention-based DRL for managing complex scheduling scenarios. Although specific datasets and detailed working models are not explicitly discussed, the approach demonstrates clear superiority in large-scale scenarios, reinforcing the significance of WSNs and reinforcement leaning in disaster response efforts. In a related study (*Wang & Lin, 2024*), DRL is applied to develop a drone system for optimal data collection from smart meters in a smart city environment. The research underscores the importance of regular re-training using real-world wireless transmission data to maintain system effectiveness. While no specific datasets or DRL models are detailed, the study emphasizes the efficiency gains achieved through DRL-based drone training, demonstrating the practical value of WSNs in smart city applications and highlighting the critical role of IoT devices, particularly smart meters, in municipal governance. Furthermore, the influence of data aggregation over wireless sensor networks is explored (*Akkaya, Demirbas & Aygun, 2008*), providing valuable insights into energy-efficient techniques for WSNs. Recognizing the increasing importance of energy-saving strategies, the research examines the trade-offs between data aggregation and key performance metrics such as energy consumption, latency, efficiency, fault tolerance, and security. By evaluating existing data aggregation methods, the authors underscore the importance of optimizing WSN performance. The study also stresses the need for collaboration between data management and networking teams, identifies future research directions, and highlights the pivotal role of data aggregation in addressing various networking challenges within WSNs.

Real-time scheduling for WirelessHART networks (*Saifullah et al., 2010*) represents a significant advancement in the field of WSNs. The research paper addresses the challenges of real-time data transmission in industrial environments, emphasizing the importance of effective scheduling within WirelessHART's centralized architecture. The authors establish that the scheduling problem is NP-hard and introduce both optimal and heuristic-based algorithms, demonstrating the critical role of these techniques in meeting end-to-end scheduling requirements. The practical implementation of the Conflict-aware Least Laxity First (C-LLF) approach, tested on both randomized and actual topologies, highlights its effectiveness in enhancing reliability and outperforming widely used real-time scheduling algorithms in wireless HART networks. Additionally, another study (*Amer et al., 2024*) contributes to energy efficiency and trajectory planning for limited multi-UAV assisted data collection in WSNs. The research discusses the challenges of transmitting data from remote WSNs to data centers and proposes a multi-UAV system to expand network coverage. The proposed method optimizes system costs and energy consumption by considering communications power, UAV mission duration, and memory constraints. The study's

focus on energy conservation and trajectory optimization underscores the critical of WSNs and UAV-assisted data gathering. Innovative approaches, such as triangulation-based K-means clustering and the Gaining-Sharing Knowledge (GSK) optimizing algorithm, demonstrate notable progress in addressing practical challenges. Moreover, *Wang & Sheng (2016)* tackles the challenges of real-time scheduling in WSNs by introducing an improved dynamic priority packet scheduling method. This approach assigns three priority queues to each sensor node, aiming to minimize node-to-node communication latency and real-time packet starvation. The technique surpasses traditional First-Come-First-Served (FCFS) and dynamic multilevel queue scheduling methods, providing its relevance in time-sensitive applications such as military surveillance and environmental monitoring. However, the review lacks specific details regarding the reinforcement learning or Deep Q-Learning models used, the datasets employed, and a thorough analysis of the approach's advantages and limitations.

The critical challenge addressed in *Zhang et al. (2015)* is the identification of events in WSNs consisting of randomly distributed spatial sensors. The proposed technique frames the detection problem as a binary hypothesis testing scenario, considering both Poisson point process and Binomial point process random deployments. The algorithms employ Askey-orthogonal polynomials for series expansion to compute the marginal probability density, offering novel framework for selecting the most appropriate expansion method based on specific system characteristics. Utilizing Monte Carlo simulations, the study validates the effectiveness of the framework and evaluates the impact of various factors on detection performance, as illustrated through receiver operating curves (ROC). In contribution to the Internet of Things (IoT) field, another study develops a DQN-based packet scheduling technique (*Fu & Kim, 2023*) for wireless sensor nodes. Recognizing the critical importance of energy-efficient resource scheduling in the IoT era, the research dynamically adjusts connection intervals and packet transmission counts across different IoT devices. Experimental results demonstrate the algorithm's effectiveness in enhancing energy efficiency and adapting to the time-varying nature of network settings. The paper further investigates DQN scheduler policies (*Luo et al., 2023*; *Wang, Chen & Li, 2023*; *Devarajan et al., 2023*; *Godfrey et al., 2023*) to gain deeper insights into optimal packet scheduling strategies. Findings indicate that the proposed scheduling algorithm significantly extends network lifetime compared to existing methods, while maintaining quality of service (QoS) under dynamic network conditions.

Previous analyses emphasize the critical role of WSNs across a wide range of applications, particularly real-time data collection and transmission. Reinforcement learning, especially DQL, has emerged as a pivotal tool for enhancing WSN performance (*Liu, 2023*; *Mohammadi & Shirmohammadi, 2023*; *Prabhu et al., 2023*; *Guo, Chen & Li, 2023*; *Pramod et al., 2023*; *Peng et al., 2023*). These models are essential for enabling WSNs to dynamically adapt to challenges such as limited energy resources and changing network conditions. The application of DQL is particularly notable in event detection and packet scheduling, where series expansion techniques and dynamic adjustments demonstrate its effectiveness. The datasets employed in these studies are largely context-specific, tailored to event detection scenarios and IoT environments. While the advantages of these approaches

include improved efficiency, extended network lifetime, and enhanced quality of service, potential drawbacks include computational complexity and resource overhead.

The above analyses collectively underscore the rapidly evolving landscape of WSNs and the promising role of reinforcement learning in their optimization. Accordingly, for our model, we conducted reinforcement learning-driven WSN-DQL simulations for signal transmission.

## BACKGROUND

In the field of enhancing real-time data transmission scheduling, the incorporation of reinforcement learning (RL) has been recognized as a crucial approach. RL, a subset of machine learning, focuses on decision-making processes by enabling an agent to learn optimum behaviours through interactions with its environment. This strategy is particularly effective in WSNs, where the dynamic and resource-constrained nature demands intelligent methods for efficient data transmission.

### Role of reinforcement learning

RL plays a critical role in enhancing real-time data transmission scheduling in WSNs. RL is a machine learning paradigm in which an agent learns to make decisions by interacting with its environment and receiving feedback in the form of rewards or penalties. In the context of WSNs, RL can dynamically adjust transmission schedules in response to changing network conditions, thereby improving efficiency and reliability. The RL framework comprise an agent, a state space, an action space, a policy, and a reward function. The agent makes decisions (actions) to maximize accumulated rewards over time, while the policy defines the strategy for selecting actions in various states. The reward function evaluates the desirability of the agent's behaviour.

$$Q(s,a) = (1 - \propto).Q(s,a) + \propto \cdot \left[ r + \gamma \cdot \max_a Q\left(s^{'}, a\right) \right] \tag{1}$$

Here, $Q(s,a)$ quantifies the appropriateness of actions $a$ in state $s$, $r$ is the immediate reward, $\propto$ is the learning rate, $\gamma$ isthe discount factor, and $s^{'}$ is the next state.

### Wireless sensor networks in data transmission

WSNs serve as a backbone for real-time data transmission, enabling seamless communication sensor nodes. Efficient data transfer in WSNs is crucial for applications such as environmental monitoring, healthcare, and smart cities. Medium access control (MAC) and time division multiple access (TDMA) are two fundamental components in the management of wireless network data transmission.

#### *Medium access control*

MAC protocols form the foundation for managing how nodes in a WSN access the communication medium and transmit data (*Bouazzi et al., 2021*). In the realm of real-time data transmission, the efficiency of MAC protocols is critical for preventing collisions and ensuring timely data delivery.

Carrier sense multiple access with collision avoidance (CSMA/CA) is a widely used MAC technology in wireless sensor networks. This protocol employs a contention-based

approach, wherein nodes first assess the availability of the communication channel before initiating transmission. The fundamental objective is to minimize the likelihood of collisions, which can significantly impact the reliability of data delivery.

The probability of successful transmission $P_{Success}$ in a CSMA/CA-based MAC protocol is determined by the ratio of the data packet size to the channel capacity:

$$P_{\text{Success}} = \frac{\text{Data Packet Size}}{\text{Channel Capacity}} \qquad (2)$$

Efficient MAC protocols improve WSN performance by enabling real-time data transfer.

### Time division multiple access

Time division multiple access (TDMA) (*Merhi, Elgamel & Bayoumi, 2009*) is a scheduling approach in which the communication channel is divided into time slots, with each node assigned specific time periods for data transmission. TDMA helps to minimize collisions and ensure fair channel access. The transmission time for each node is determined as follows:

$$T_{\text{slot}} = \frac{\text{Total Time}}{\text{Number of Nodes}} \qquad (3)$$

TDMA is highly effective in scenarios where precise timing is essential, such as in real-time data transfer applications. It ensures predictable and controlled access to the communication channel, thereby enhancing the reliability of WSNs.

In conclusion, RL, MAC and TDMA are fundamental to Wireless Sensor Networks, ensuring efficient and real-time data transmission scheduling. The integration of RL algorithms with MAC and TDMA protocols (*Karthick, 2023*; *Ahmed et al., 2023*; *Debadarshini et al., 2023*; *Raut & Khandait, 2023*; *Chittapragada et al., 2023*; *Popovici & Stangaciu, 2023*) significantly enhances adaptability by dynamically optimizing transmission strategies in response to evolving network conditions.

## Deep Q-learning fundamentals

DQL stands at the forefront of reinforcement learning methodologies, transforming the optimization of decision-making processes in dynamic environments. Fundamentally, DQL integrates the capabilities of deep neural networks and Q-Learning, enabling agents to learn optimum strategies through interaction with their surroundings. In the context of real-time data transmission scheduling, DQL offers a robust framework for dynamically allocating resources in WSNs, ensuring timely and efficient communication between sensor nodes.

In the DQN network framework, the neural network functions as a supervised learning mechanism for WSNs. The general approach constructs two Q networks, where the target $Q$ value defines the loss function. The input pool provides training samples, and the parameters W and the bias b are updated using the stochastic gradient descent method (SGD), which also calculates the gradient after estimating $Q$ value. This process illustrates the structure of the DQN network model.

Figure 1 illustrates two networks constructed with the same topology but exhibiting different characteristics. One network predicts and calculates the $Q$ value is using the most

recent parameters, while the other updates the $Q$ value based on parameters from an earlier point in time. This approach enhances the algorithm's reliability more reliable, ensures greater stability of the target $Q$-value over a period, reduces the correlation between the current and target $Q$ values. The experience pool, also known as experience replay, serves not only to provide training examples but also to address the issue of data correlation. At the outset of the training process, a memory bank is initialized. The experience pool stores the current state, action, reward, and the resulting state in the next time slot following the execution of an action. During network training, a specified amount of memory is utilized, with mini-batches randomly sampled from the experience pool. Furthermore, once the experience pool reaches capacity, new experiences overwrite the oldest ones, disrupting the original data sequence and further minimizing information correlation.

# WSN NETWORK MODEL

This research introduces a real-time data transmission scheduling method based on deep recurrent neural networks (RNNs) to address the challenge of concurrent data transmission in WSNs. The proposed method considers key parameters such as the remaining time before the deadline, the number of remaining hops left, and the unallocated time slots for nodes during data transmission. It also defines an action selection strategy and a reward function for the deep RNN which guide the information transmission process based on system state data (*Gu et al., 2023*; *Wu et al., 2023*; *Chauhan & Kumar, 2023*).

The key contributions of the study are as follows:

i. Comprehensive consideration of transmission factors: The proposed method incorporates all critical factors influencing data transmission, including remaining cut off time number of remaining hops left, and nodes without allocated time slots. This holistic approach enhances the efficiency and adaptability of WSN scheduling.

ii. Recurrent attention-deep Q learning for action selection: The action selection process is guided by DQL integrated with attention and residual mechanisms. The study employs bidirectional long short-term memory (BiLSTM) to manage complex, large-scale system states-related disaster recovery scenarios. BiLSTM effectively predicts and establishes the state-action mapping, thereby improving overall system robustness.

iii. The methodology enhances scheduling decisions by defining a robust reward function and strategy for deep residual Q learning. It incorporates optimal node constraints and an effective penalty mechanism, ensuring a more intelligent and efficient of real-time data transmission process.

WSN network structure as shown in Fig. 2 was detailed below:

## Networking structure

WSN features a predefined topology and consists of M sensor nodes and one base station node. The sensor nodes are responsible for data generation and transmission tasks, while the base station functions as the destination node. Sensor nodes periodically generate data with specific deadlines and transmit it to the base station through inter-node communication within designated time slots.

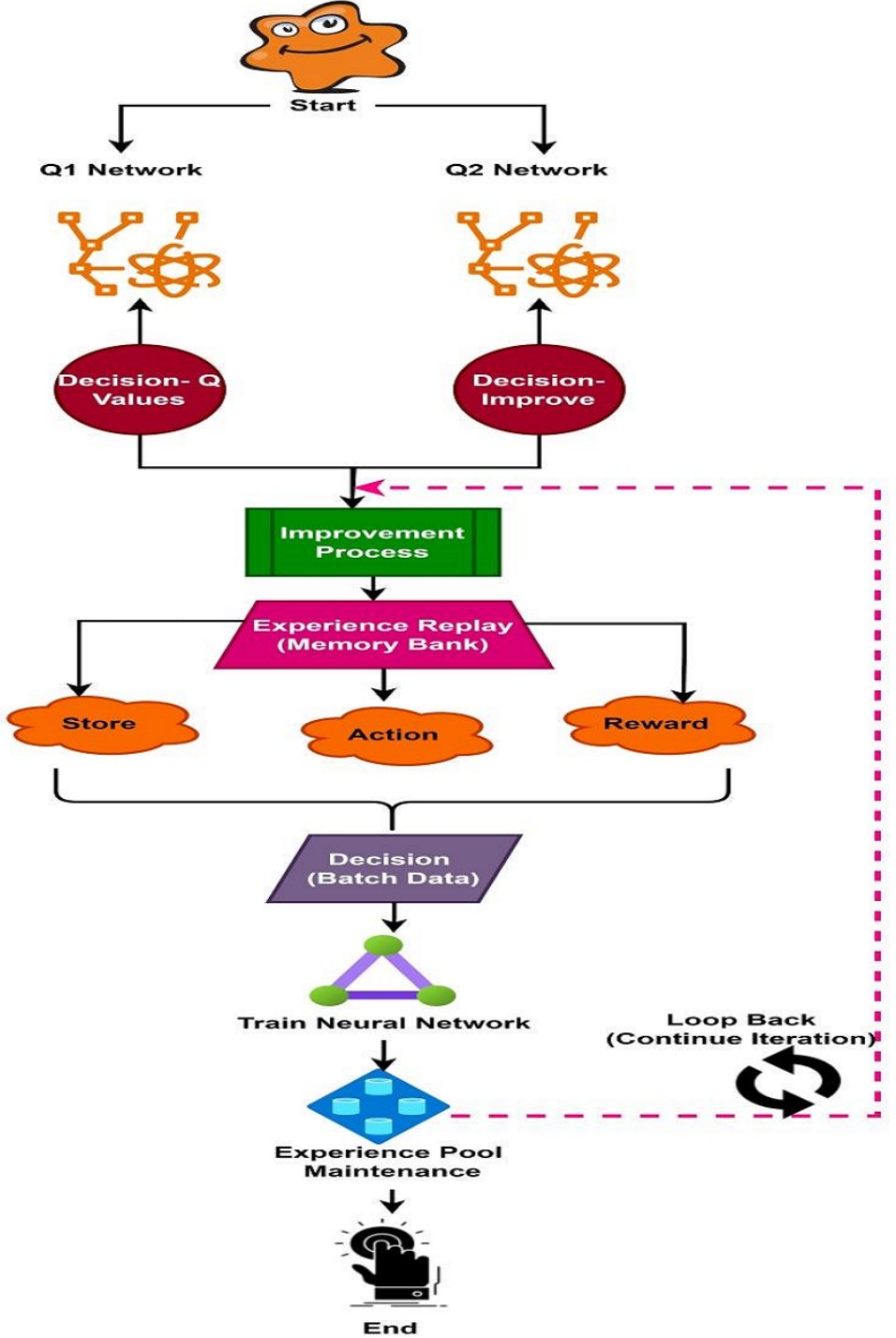

**Figure 1** Workflow of DQL network.

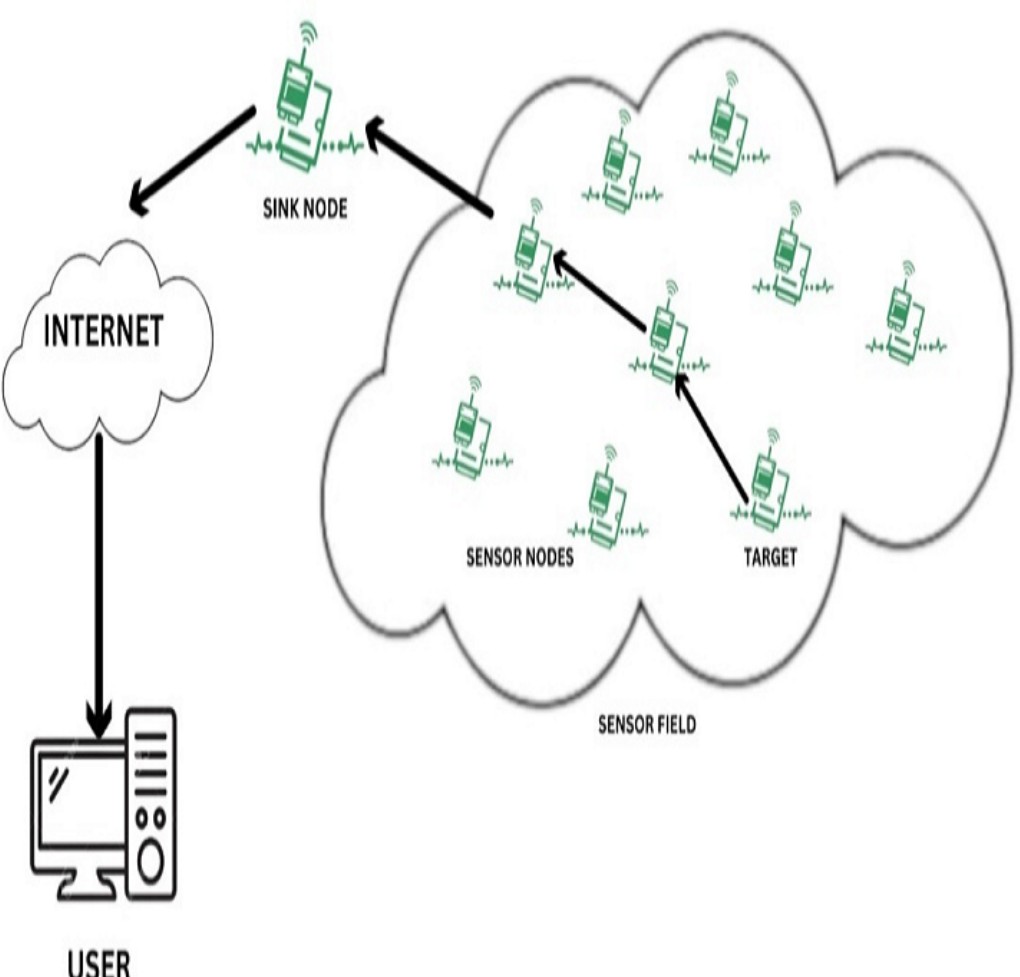

**Figure 2** WSN network structure.

## Packet representation of data

The data packet generated by the i-th node, denoted as $P_i$, is defined as: $P_i = T_i, H_i, D_i,$ $\phi_i$. In this case, T i represents the packet generation interval, $D_i$ is the packet transmission deadlines $\phi_i$ indicates the routing path of the packet, and $H_i$ denotes the total number of hops from the source node to the sink node. The parameters $T_i$ and $D_i$ are expressed in units of time slots. Typically, to ensure sufficient time for successful transmission $T_i$ must be greater than $H_i$.

## Packet properties at time slot

At any given time slot t, the data packet is characterized by three key attributes: $C_i$, $h_i$, $t_i$. $(C_i \epsilon \phi_i)$, represents the current node location of the data packet and $h_i$ indicates the number of remaining hops in the transmission process. $(0 < t_i \leq T_i)$ denotes the remaining time to meet the transmission deadline.

### Method of transmission

Transmission is considered feasible if $t_{i.} > h_{i.}$. When a time slot is reserved for transmission, both the remaining time and the number of hops are reduced by one, facilitating progress to the next node. If transmission does not occur, only the remaining time $t_{i.}$ is reduced by one. When $t_i = 0$, the node generates a new data packet and enters a waiting phase for transmission scheduling, initializing with $t_i = T_{i.}$ and $h_i = H_{i.}$

The proposed model is based on the following realistic assumption to guide the practical design and implementation of WSNs:

i.  Exclusive transmission capability: sensor nodes are not capable of simultaneous data transmission and reception, nor can they receive from multiple nodes concurrently.

iii.  Single data packet selection: to ensure effective resource use, nodes receiving multiple data packets can only choose one packet to transmit during each time period.

iii.  Strict cut-off time for data: in order to preserve synchronization, data created by source nodes on a regular basis are subject to a stringent cut-off time constraint that corresponds with the data creation cycle.

iv.  Immediate discard for exceeded deadlines: to avoid transmitting out-of-date or unnecessary information, a data packet is immediately deleted if its transmission deadline is exceeded.

v.  Effect of physical elements on communication: transmission power, encoding style, and modulation scheme are examples of physical elements that affect the likelihood of a successful wireless communication transmission.

vi.  Probability of performance in transmission: the modeling method is made simpler while admitting the unpredictability in wireless communication performance by assuming that if data is organized for transmission, the likelihood of success is 1.

vii.  Time constraints are taken into account: it is acknowledged that data transmission is time-sensitive, guaranteeing prompt delivery of information within designated deadlines.

viii.  Optimization for resource efficiency: by maximizing resource allocation and usage within the bounds of realistic WSN installations, the assumptions seek to enhance network performance and data transmission efficiency.

## DEEP Q-NETWORK MODEL DATA TRANSMISSION SCHEDULING

In the context of real-time data transmission scheduling, DQL provides a comprehensive framework for dynamically allocating resources in WSNs (*Zhang et al., 2022*; *Luo, 2020*; *Xiong, He & Lu, 2025*; *Zhang et al., 2025*), assuring timely and efficient communication between sensor nodes. The following section clearly shows the methodology of the proposed recurrent attention-deep Q learning with optimal node constraints and effective penalty based WSN scheduling RA-DQL-ONC&EP.

## State representation

The state in your system should include information that the agent (the deep RNN) can use to make scheduling decisions. A suitable state representation in the existing model for data transmission scheduling in WSNs

1. **Buffer status:** information about the data buffer status of each sensor node. For example, the number of packets waiting to be transmitted.
2. **Network conditions:** data related to the current network conditions, such as the channel occupancy, signal strength, interference, and available bandwidth.
3. **Historical schedules:** information about past scheduling decisions and their outcomes, including success rates and energy consumption.
4. **Data urgency:** priority of data transmission

This state information at time step t can be represented as S(t). In this work different kind information about each sensor node is represented by the state. Those are described as follows.

### A. Number of hops

A fundamental characteristic of a node in a WSN is the number of hops, or intermediate relay nodes, required to transmit data along the communication path between the originating node and the target node. Due to the limited signal amplification capabilities inherent in sensor nodes, direct communication is typically impractical. Consequently, multi-hop communication becomes indispensable for enabling long-range data transmission within WSNs. Depending on the network design, data can be transmitted using either a few long distance hops or several short-distance hops. Each transmission strategy long hop or short hop offers distinct advantages. Long-hop transmission minimizes the number of participating nodes, thereby reducing routing complexity and associated overhead costs. Conversely, short-hop transmission is more energy-efficient, as the energy required for wireless transmission increases with distance. Therefore, selecting the appropriate hop strategy involves balancing trade-offs between transmission cost and energy consumption.

### B. Packet life time

All data packets at nodes should have limited lifetime. The lifetime represents its actual time period from source to destination.

### C. Node degree

Node degree is common parameter used in all WSN scheduling as well as routing. It estimated for each node based on the number of one-hop neighbours connected to it. If a node has a higher node degree, it can transmit the data to more neighbours, which likely increases the success rate of transmission. Node degree is represented as $N_{deg}$ and can be expressed follows:

$$N_{deg}(N_i) = \| N_j \| \; where \; j \in \text{one hop neighbour of } N_i \tag{4}$$

### D. Node energy and average energy of neighbour nodes

Due to limited energy resources and distances between nodes, communication between sensor nodes and sink occurs *via* a multi-hop model. Each node forwards the collected measurements to neighbouring nodes, which subsequently relay them further. Monitoring the energy consumption of active sensor nodes allows applications and routing protocols to make intelligent, data-driven decisions, thereby enhancing the overall operational longevity of the sensor network.

In this module, the current node's energy and the average energy derived from its neighbouring nodes are maintained as state parameters. Information such as residual energy, node location, and the type and size of packets from one-hop neighbouring nodes is used to calculate the average energy of neighbouring nodes. Each node records the information of its neighbour upon receiving a packet. Once neighbouring nodes are identified, the node calculates their average energy as follows

$$E_{avg}(N_i) = \frac{\sum_{j \in Neg(N_i)} Energy(N_j)}{\| Neg(N_i) \|} \tag{5}$$

where $Eavg(i)$ is the average energy of neighbouring node i, $Neg(N_i)$ is the set of neighbour nodes, $\| Neg(N_i) \|$ is the number of neighbors, $j$ is the neighbour node of node $i$ in Eq. (5)

### E. Node local balancing and spatial coverage parameters

In WSN, each node may have a varying number of neighbours and may be distributed in different spatial patterns. Therefore, determining balanced neighbour coverage is a key factor in ensuring efficient data transmission. This metric indicated whether nodes are uniformly distributed near the source node or whether there is significant variation in their spatial arrangement. From this value, it is possible to determine if certain nodes are closer to the boundary of the coverage area, an aspect that enhances the likelihood of identifying the next hop node and helps avoid communication voids. This module extracts two values to characterize the distribution of the neighbouring nodes as local balancing coverage (LBC) local spatial coverage (LSC).

As illustrated in Fig. 3, the neighbour nodes of n1 and n2 fall within the transmission range r. In terms of local balancing coverage, node n1 exhibits better distribution than node n2, as its neighbours are positioned at relatively equal distances. In contrast, node n2 has neighbours at varying distances, indicating an uneven distribution. The local balancing coverage for a node '$i$' is calculated as follows: firstly, the distance between each neighbour node '$j$' from node '$i$' is computed and stored in a list called $DList$. Then, for each entry in DList, the average distance deviation ($ADist_j$) is calculated and compiled into a list $DList$ as $ADList$. Finally, the LBC is derived as the ratio of the difference between the maximum and minimum values in $ADList$ to the transmission range $r$.

$$Dist_j = \sqrt{(x_i - x_j)^2 + (y_i - y_j)^2}, where \; j \leftarrow Neg(N_i) \tag{6}$$

$$ADist_j = abs(Dist_j - Mean(DList)) \tag{7}$$

$$LBC(N_i) = \frac{(\max(ADList) - \min(ADList))}{r}. \tag{8}$$

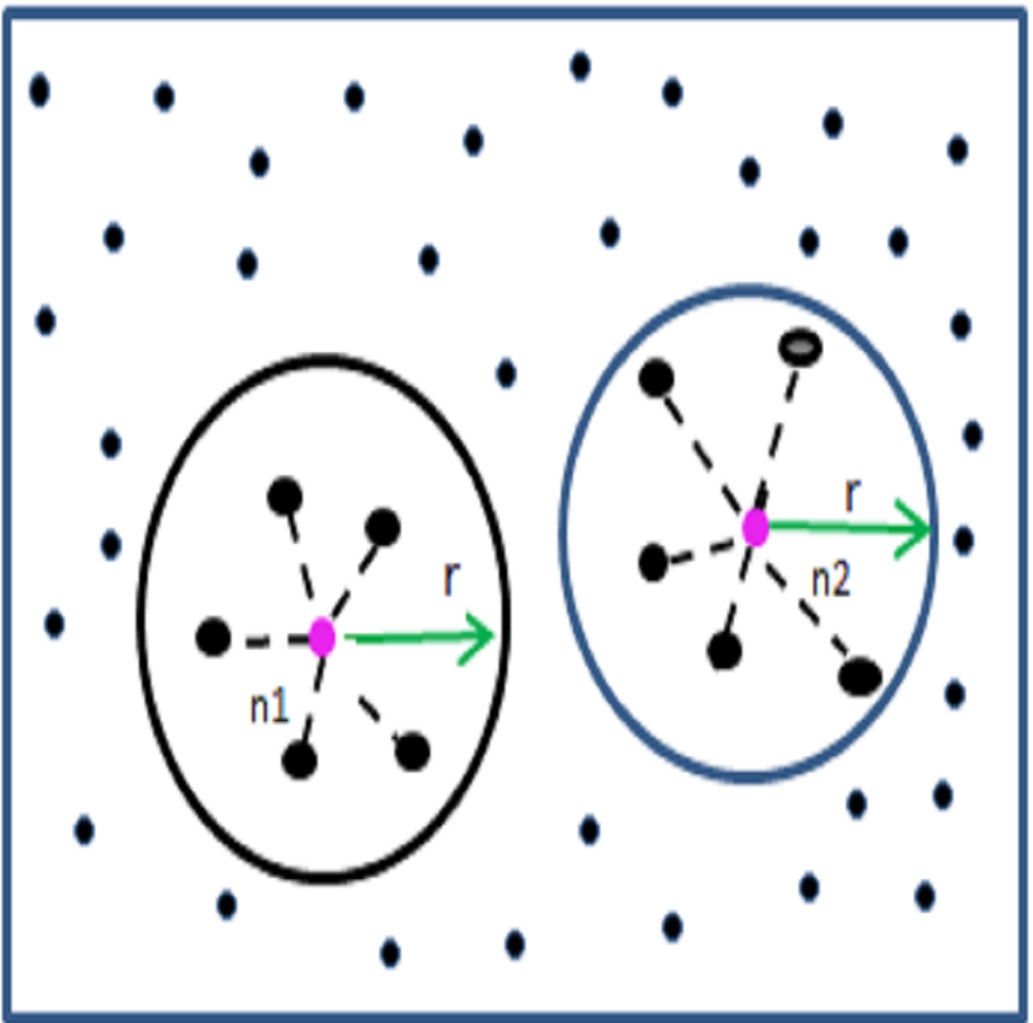

**Figure 3** **Example for LBC and LSC estimation.**

Let us consider the node n1 has five neighbour nodes in the distance of 105, 100, 110, 115 and 90 from the position of n1 within the transmission range 200 and its average is 104. Then for each neighbour the $ADist_j$ is estimated using the Eq. (7) and from that the $LBC(n1)$ is estimated using the Eq. (8) which is 29.9. This value indicates that the nodes are covered in balanced range from n1 compared to the $LBC(n2)$ which is around 50, where the neighbour nodes are in the distance 190, 180, 90, 190 and 60 from n2.

Similarly the local spatial coverage is estimated using the following Eq. (9). The detected $LSC(n_1)$ is 0.52 which shows the nodes are more or less in half radius from its coverage, whereas the $LSC(n_{21})$ is 0.71 it reflects most of the nodes are nearer to boundary region. It is clearly noted in the Fig. 3.

$$LSC(N_i) = Mean(DList)/r. \tag{9}$$

### F. Joined link coverage score

In this study, an additional metric joined link coverage score is introduced. This score evaluates the cumulative interconnectivity of each neighbouring node with other neighbouring nodes, based on the number of shared one-hop neighbours of node $i$. Accordingly, it is termed joined link coverage. The computation begins by determining the reciprocal of the number of common neighbours of each one hop neighbour and the others. The average of these values is then calculated to yield the JLCS. The estimation is presented in Eq. (10). Nodes with more interconnected neighbours receive a lower score compared to those whose neighbours are parsley connected. This metric is instrumental the scheduling overhead in the network.

$$JLCS(N_i) = mean\left(\frac{1}{\|\,Neg\,(N_j \cap Neg\,(N_k))\,\|}\right) \tag{10}$$

where $j, k \leftarrow Neg(Ni)$

### G. Hop waited score

Normally the packet adjacency score of a node where at presents is estimated with the help of the number of hops to travel and the time to live or lifetime of the packet which is estimated as follows:

$$US(N_i) = \left(\frac{\text{Number of Hops to Travel}}{\text{Life Time of the Packet}}\right). \tag{11}$$

This urgency score serves as one of the input parameters for the DQN model. Once calculated for the current state, the algorithm compares all the urgency scores of all nodes. In traditional models, the node with the highest urgency score typically receives the highest scheduling priority. However, this algorithm introduces a more nuanced approach by evaluating whether other nodes are at risk of missing a valuable scheduling opportunity. The procedure is described as follows: $US(N_k)$ represent the highest urgency score among all 'n' nodes in the network. The algorithm compares this value with the urgency scores of the remaining nodes. If any node's urgency score falls within a specific threshold of the maximum urgency score, that node is also considered for scheduling. Subsequently, the hop-ratio of the considered node is calculated relative to the hop count of node 'k'. If this ratio is less than 60%, the urgency score is further increased by applying a hop-based weighting factor.

$If\,(US(N_i) - US(N_m)) \leq 0.2\;then:$

$$HDev = \left(\frac{\text{Number of Hops to Travel from } N_m}{\text{Number of Hops to Travel from } N_i}\right)$$

$If\;HDev \leq 0.6\;then:$

node $m'$ selected for urgency score adjustment

$$US(N_m) = US(N_m) + (US(N_i) * (1 - HDev)). \tag{12}$$

From this itself, the Hop weighted Missing Score ($HWMS$) is also estimated, which is used for the penalty estimation of DQN. The hop weighted missing score is estimated after the

action state as, if the current state is not allotted even though it attains hop based weight score means 30% of its urgency score is assigned as the hop weighted missing score.

In this work reward and penalty is estimated with the constraints values and scores.

$$Reward = w1 * Reward = w1 \times US + w2 \times \frac{1}{Eavg} + w3 \times \frac{1}{LBC} \tag{13}$$

where w1 + w2 + w3 = 1

$$Penalty = w1 \times \frac{1}{JLCS} + w2 \times \frac{1}{HWMS}. \tag{14}$$

## WSN-DQL-BiLSTM attention-based architecture for data transmission scheduling

This section presents the wireless sensor network-deep Q learning bidirectional long short-term memory (WSN-DQL-BiLSTM) WSN-DQL-BiLSTM attention-based architecture, developed for real-time data transmission in WSNs, context-driven attention layers combined with bidirectional sequence, the architecture is designed to effectively capture temporal dependencies and emphasize relevant features within the incoming data. The model enhances the network's ability to make informed scheduling decisions by combining attention models with dense layers. This approach enables WSNs to deliver reliable and efficient real-time data transmission, allowing them to dynamically adapt and optimize transmission schedules in response to changing network conditions.

Figure 4 illustrates the BiLSTM architecture for WSN-DQL. The architecture comprises several layers and operations aimed at extracting meaningful features from the input data and facilitating decision-making in real-time data transmission scheduling. Initially, three inputs (IP-1, IP-2, IP-3) are fed into BiLSTM layers (Bi-1, Bi-2, Bi-3) separately, each with a size of 128. BiLSTM layers are capable of capturing both forward and backward temporal dependencies in the input sequences, making them well-suited for sequential data processing tasks in WSN-DQL. Subsequently, three attention models (AT1, AT2, AT3) are applied to obtain weight matrices, quantity, attention weight, and each time series attention matrix, which help to emphasize relevant features and suppress noise in the input data.

After applying attention models, Avgpool (AP) is individually applied to each attention model output (AT1, AT2, AT3). The resulting values from AP1 and AP2 are concatenated (C1) and passed through a dense layer (C1_D1) with a size of 128. Similarly, the output from AP3 is fed into another dense layer (Bi-3_D2). Following the dense layers, addition (A) and subtraction (S) operations are performed separately on the outputs of dense layers C1_D1 and Bi-3_D2. These operations allow for combining and comparing the features extracted from different parts of the input data.

Next, dense layers (D3 and D4) with a size of 64 are applied to the results of addition (A) and subtraction (S). The outputs from these dense layers are concatenated (C2), followed by additional dense layers (D5, D6, D7) with sizes 64, 32, and 2, respectively. These dense layers help in further feature extraction and dimensionality reduction, enabling the model to capture essential information for decision-making. Finally, a softmax activation function is employs on the output of dense layer (D7) to produce the final outcome of

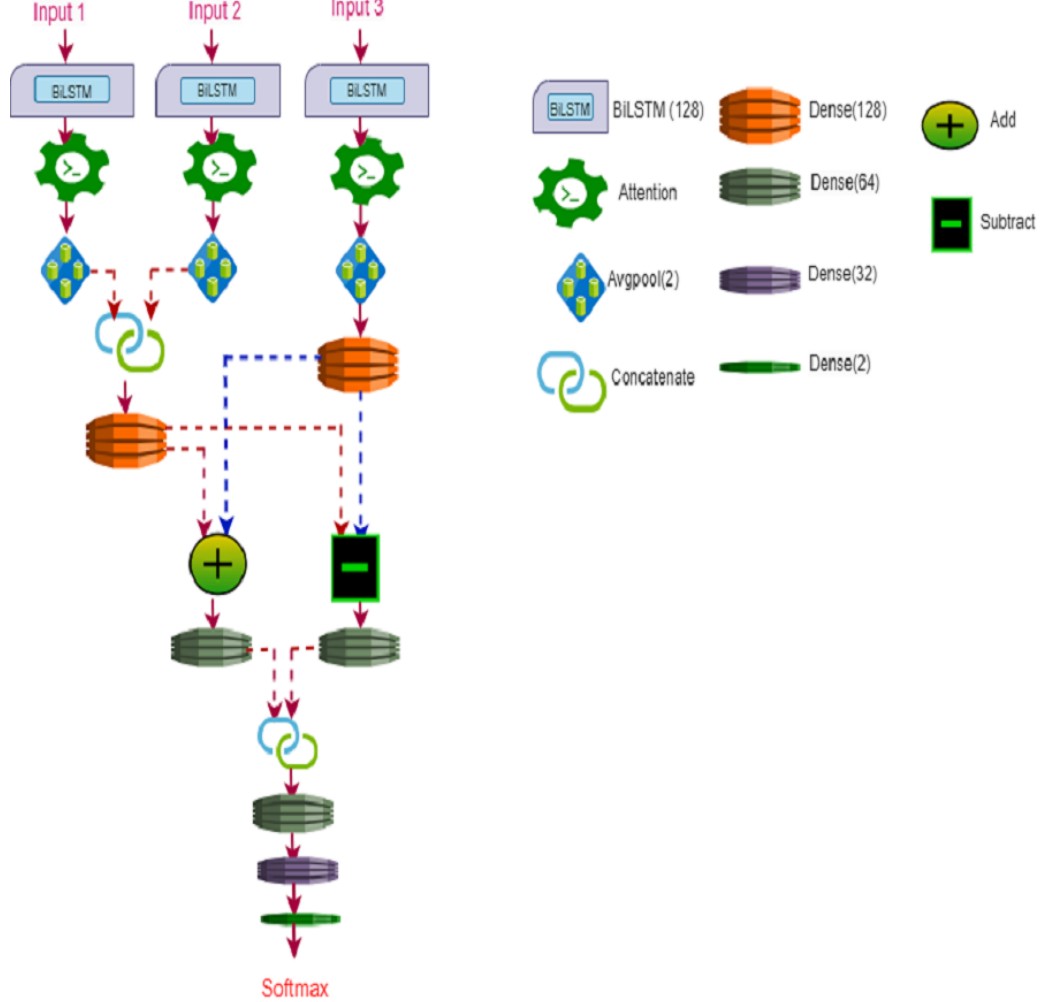

**Figure 4  WSN-DQL's BiLSTM architecture.**

WSN-DQL-BiLSTM, which represents the probabilities of different actions or decisions in the context of real-time data transmission scheduling in WSNs.

The operations described above can be represented as follows:

$$Bi_{out_i} = Bi(IP_i) \tag{15}$$

$$AT_i = AT\left(Bi_{out_i}\right) \tag{16}$$

$$AP_i = AP(AT_i) \tag{17}$$

$$C_{1D_1} = ense(Concat(AP_1, AP_2), 128) \tag{18}$$

$$Bi_{3_{D_2}} = Dense(AP_3, 128) \tag{19}$$

$$A_{D_1} = \left( C_{1_{D_1}} + Bi_{3_{D_2}} \right) \tag{20}$$

$$S_{D_2} = \left( C_{1_{D_1}} - Bi_{3_{D_2}} \right) \tag{21}$$

$$D_{Out_3} = Dense\left( A_{D_1}, 64 \right) \tag{22}$$

$$D_{Out_4} = Dense\left( S_{D_2}, 64 \right) \tag{23}$$

$$C_2 = Concat\left( D_{Out_3} \& D_{Out_4} \right) \tag{24}$$

$$D_{Out_5} = Dense(C_2, 64) \tag{25}$$

$$D_{Out_6} = Dense(D_{Out_5}, 32) \tag{26}$$

$$D_{Out_7} = Dense\left( D_{Out_6}, 32 \right) \tag{27}$$

$$Final_{Outcome} = Softmax(D_{Out_7}). \tag{28}$$

This sophisticated architecture aims to leverage the capabilities of BiLSTM and attention mechanisms for robust learning and decision-making in Wireless Sensor Networks, providing a comprehensive framework for real-time data transmission scheduling in dynamic and resource-constrained environments.

## EXPERIMENTAL RESULTS

The simulation experiment evaluates the network performance of WSN data packets under varying random deadlines for deadlines for scheduling. The simulation is implemented using the wsnsimpy Python library and conducted in a Python 3.8 environment over 10,000 timestamps. The experimental setup examines different node densities, ranging from five to 100 nodes. The performance of the proposed deep queue model is assessed using key metrics such as packet success rate, delay, energy consumption and throughput. To facilitate a comprehensive comparison, an extended time window is to analyse and compare the volume of lost packets across scenarios. Table 1 displays additional simulation settings.

The main objective of the goal of the RS-DQL technique is performance optimization through real-time data transmission scheduling in wireless sensor networks *via* deep Q-learning. It is contrasted with the traditional earliest deadline first (EDF) algorithm, which ranks data according to earliest time constraints, and the EDP algorithm, which sorts data into priority queues according to features like periodicity and urgency. Data transmission is prioritized by EDP according to remaining time and hop count, while EDF chooses non-conflicting nodes with the shortest deadline for transmission. Performance

**Table 1 Simulation parameters.**

| Parameters | Value |
|---|---|
| Learning rate | $\partial = 0.001$ |
| Discount factor | $\Gamma = 0.9$ |
| Episode or iteration | 10,000 |
| Nodes count | 5 to 100 |
| Optimizer | Adam |
| Packet cutoff time | Randomly generated for each packet |

**Table 2 Average number of lost packets for different algorithms.**

| Algorithm | EDP | RS-DQL | RA-DQL-ONC | RA-DQL-ONC&EP |
|---|---|---|---|---|
| Number of lost packets | 4,078 | 3,067 | 1,655 | 1,298 |
| Number of successfully sent packets | 10,695 | 11,706 | 13,118 | 13,475 |
| Packet loss rate | 28.22% | 21.06% | 14.12% | 12.20% |

comparisons under various conditions—like data deadlines and the number of network nodes—showcase how effective RS-DQL is at transmission schedule optimization.

Table 2 presents a comparative evaluation of four scheduling algorithms—EDP enhanced dynamic priority (EDP), RS-DQL random scheduling-deep Q learning (RS-DQL), recurrent attention-deep Q learning with optimal node constraints (RA-DQL-ONC), and recurrent attention-deep Q learning with optimal node constraints and effective penalty based WSN scheduling (RA-DQL-ONC&EP)—based on their performance metrics in a simulated WSN environment. The number of lost packets, successfully sent packets, and packet loss rates are evaluated to assess algorithm performance. Notably, RA-DQL-ONC&EP demonstrates superior performance, with the lowest packet loss rate of 12.20%, followed closely by RA-DQL-ONC at 14.12%. EDP and RA-DQL-ONC exhibit higher packet loss rates, indicating less efficient scheduling. The results suggest that incorporating deep Q-learning, attention mechanisms, optimal node constraints, and penalty-based models enhances scheduling efficacy, with RA-DQL-ONC and RA-DQL-ONC&EP showing promising outcomes.

In Fig. 5, the packet loss rate of the RA-DQL-ONC&EP algorithm is notably lower compared to RA-DQL-ONC, RS-DQL, and EDP. Specifically, it exhibits a reduction of 2.41% in packet loss rate compared to RA-DQL-ONC, showcasing the effectiveness of incorporating effective penalty-based models in optimizing data transmission scheduling. Moreover, RA-DQL-ONC&EP demonstrates substantial improvements with 11.97% less packet loss compared to RS-DQL and a significant 18.81% reduction compared to EDP. These results underscore the efficacy of the algorithm in mitigating packet loss and enhancing the reliability of wireless sensor network communication.

The RA-DQL-ONC&EP method, which has a notable success rate of 91.21%, is shown in Fig. 6 with its success rate. In comparison to previous models, this proposed method shows notable gains: its success rate beats that of RA-DQL-ONC by 2.41%, RS-DQL by 11.97%, and EDP by 18.81%. The algorithm's capacity to optimize data transmission

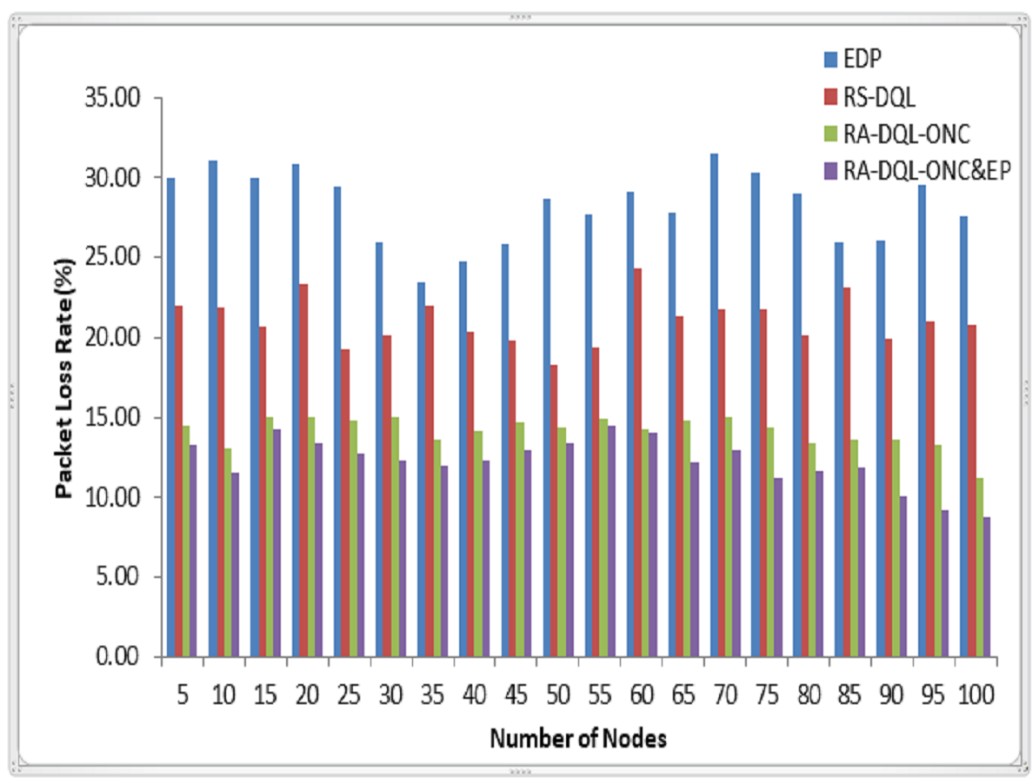

**Figure 5** Packet loss rate analysis.

scheduling in wireless sensor networks is improved by these results, which highlight the usefulness of adding optimum node constraints and penalty-based models to the recurrent attention-deep Q Learning framework.

The delay rate for the RA-DQL-ONC&EP model is 1.99 in Fig. 7. Compared to previous models, this result shows a significant improvement: it is 1.12% lower than RS-DQL, 1.4% lower than EDP, and 0.16% lower than the delay shown in RA-DQL-ONC. These findings highlight the value of using efficient penalty-based scheduling and optimum node restrictions in the RA-DQL framework, as they minimize data transmission delays in wireless sensor networks. These improvements demonstrate RA-DQL-ONC&EP's ability to reduce latency and improve network performance in practical deployment circumstances.

In Fig. 8, the energy consumption in joules (J) for node 100 is depicted for the RA-DQL-ONC&EP algorithm. It shows that RA-DQL-ONC&EP consumes 70% of the energy compared to other models. Specifically, compared to RA-DQL-ONC, it consumes 10.51% less energy, 23.11% less energy compared to RS-DQL, and 35.37% less energy compared to EDP. These findings underscore the significant energy efficiency improvements achieved by RA-DQL-ONC&EP, highlighting its effectiveness in optimizing energy consumption in wireless sensor networks.

Figure 9 illustrates the analysis of throughput for each time step in the context of the studied models. Specifically, in Fig. 9, the throughput for RA-DQL-ONC&EP is depicted as

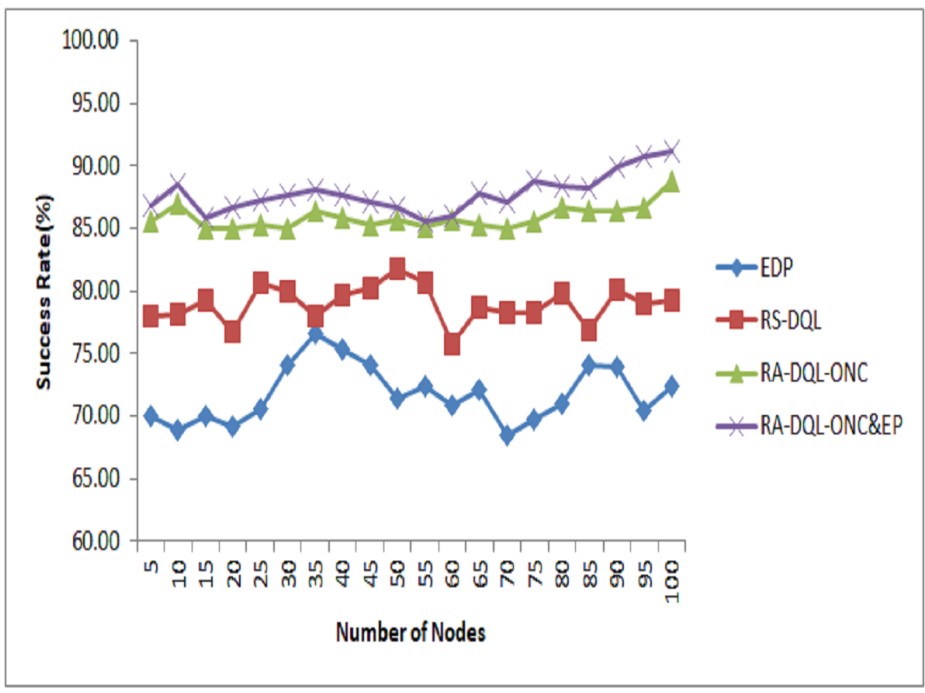

**Figure 6**  **Success rate analysis.**

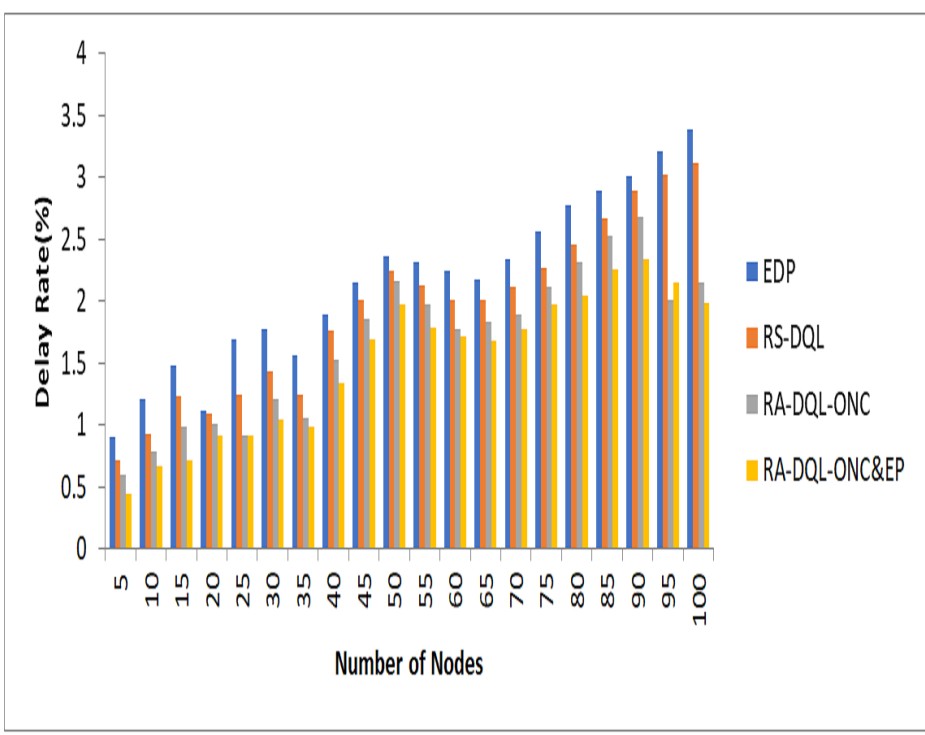

**Figure 7**  **Delay analysis.**

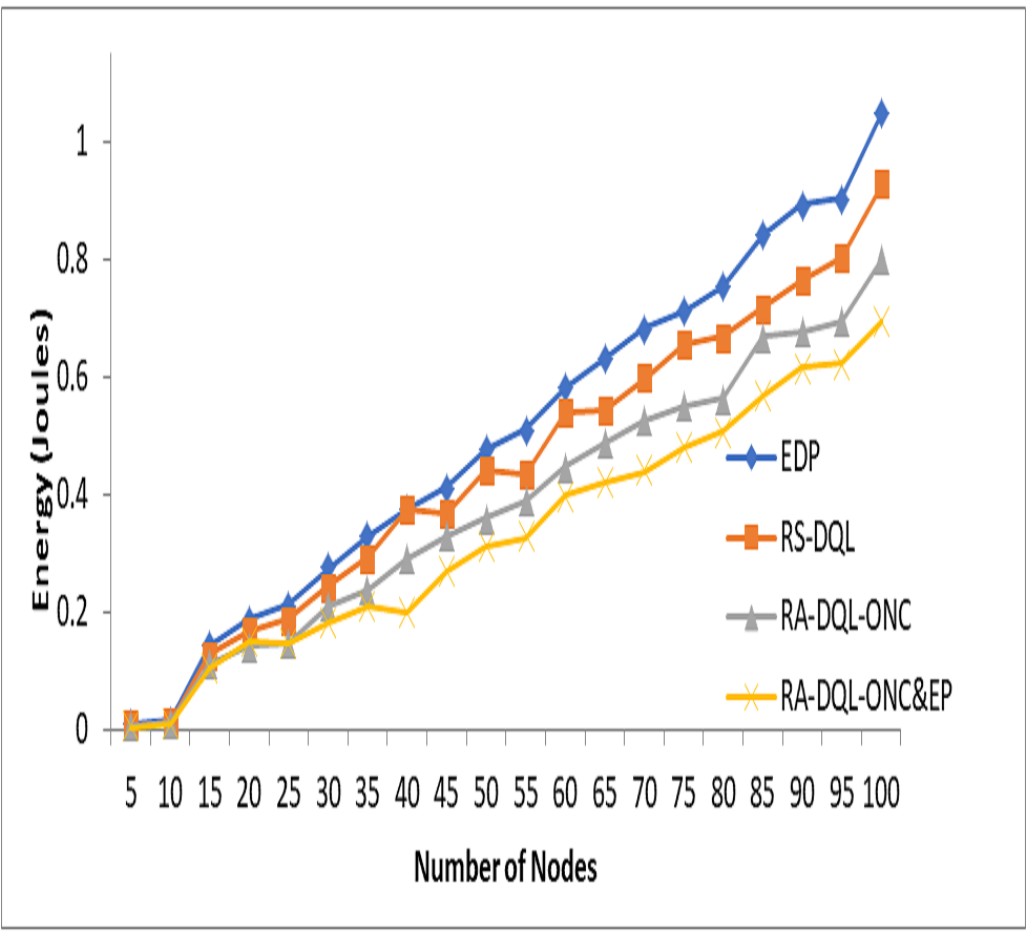

**Figure 8** Energy jules analysis.

72% out of 1,000 time steps. Comparatively, this throughput is observed to be 15% higher than the energy-efficient RA-DQL-ONC, 27% higher than RS-DQL, and 36% higher than EDP. These findings suggest that while RA-DQL-ONC&EP exhibits substantial throughput, it falls short in comparison to other models in terms of maximizing data transmission efficiency over the simulated time steps.

## CONCLUSION

The RA-DQL-ONC&EP algorithm presents a viable solution for enhancing data transmission scheduling in WSNs. This approach integrates optimal node constraints, a penalty-based model, and recurrent attention mechanisms to maximize scheduling efficiency while accounting for energy consumption and network interference. The technique has proven to significantly increase network performance indicators, such as success rate, delay rate, and energy efficiency, through simulated trials. These results highlight its potential to help WSNs overcome obstacles and improve their resilience and dependability in real-world deployments. Future research may explore several

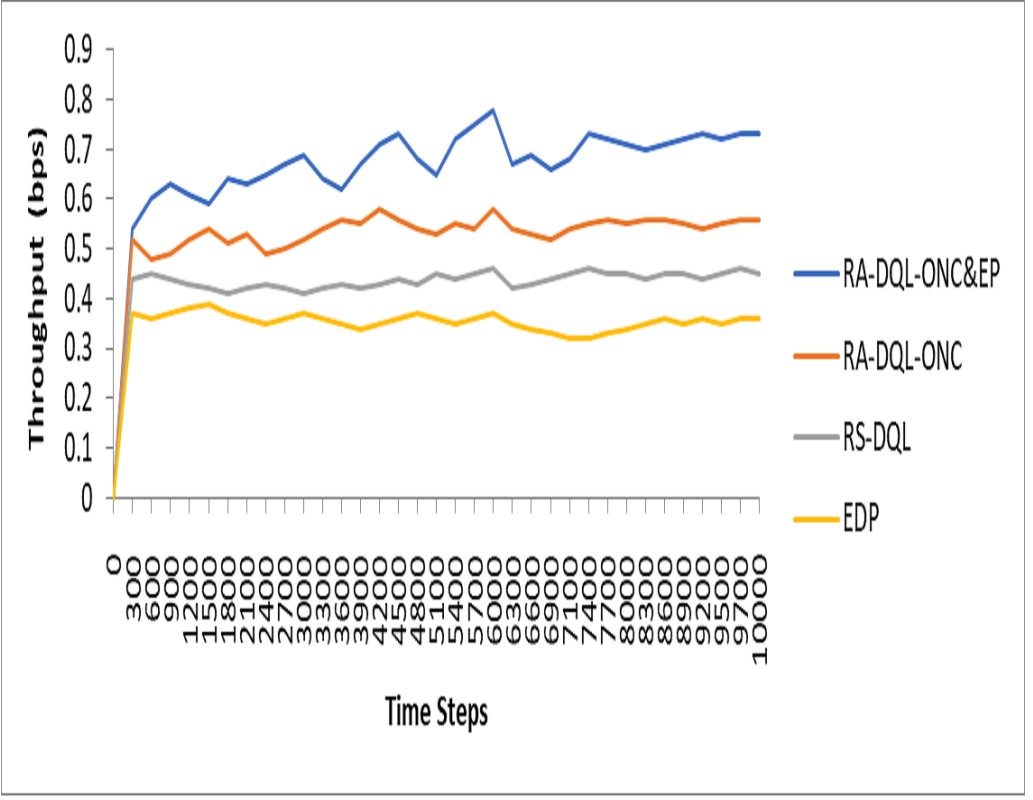

**Figure 9** Analysis of throughput for each time steps.

directions. Firstly, further optimization and fine-tuning of algorithm parameters could lead to enhanced performance and better adaptability to diverse network conditions and application requirements. Secondly, investigating the scalability of the algorithm to larger and more complex WSNs is essential for practical deployment. Lastly, assessing its robustness against various network dynamics—such as node failures, environmental changes, and fluctuating traffic patterns—will be critical to ensuring dependable and efficient performance across a wide range of operational scenarios.

### Funding
The authors received no funding for this work.

### Competing Interests
The authors declare there are no competing interests.

## Author Contributions

- D.R. Anita Sofia Liz conceived and designed the experiments, performed the experiments, performed the computation work, prepared figures and/or tables, and approved the final draft.
- Yesubai Rubavathi C analyzed the data, authored or reviewed drafts of the article, and approved the final draft.

## Data Availability

The data were generated using the wsnsimpy Python library as simulated data:

1 GitHub Repository: https://github.com/saraHossein/wsn-simpy-

2 PyPI Package: https://pypi.org/project/wsnsimpy/.

The code is available in the Supplementary File.

## Supplemental Information

Supplemental information for this article can be found online at http://dx.doi.org/10.7717/peerj-cs.2950#supplemental-information.

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
