# Peer review of "Enhanced recurrent attention-deep Q learning with optimal node constrains and effective penalty based model for data transmission scheduling on wireless sensor networks"

_PeerJ Computer Science, doi:10.7717/peerj-cs.2950_

## Round 0.1 · original submission · Major Revisions

Dear Authors,

The manuscript has been reviewed by two expert reviewers. Please refer to the reviewers' comments, and revise the manuscript accordingly.

Best regards,
Woorham

Reviewer 1 ·

Basic reporting

Overall, the paper is well-written, but improvements are needed to enhance the readability of the equations and figures.

Experimental design

no comment

Validity of the findings

no comment

Additional comments

The authors have proposed a novel method called RA-DQL-ONC&EP, demonstrating its potential to improve the performance of wireless sensor networks. They have also shown, using various metrics including delay, energy, and throughput, that the proposed method outperforms other existing methods. I have a few questions related to the content.

1. Could you please provide a more detailed explanation of the statement mentioned in the Introduction, "(55) An uneven clustering method equalizes energy consumption among cluster heads"?

2. Figures 7-9 present important analyses, but they are difficult to interpret. Please add units to the Y-axis. For example, the text mentions the delay rate (%), but the figures are labeled simply as "delay."

3. According to Figure 7, as the number of nodes increases, the delay for EDP and RS-DQL generally increases, whereas the delay for RA-DQL-ONC and RA-DQL-ONC&EP decreases again. What do you think is the most likely reason for this?

4. In Figure 9 (552), the throughput of RA-DQL-ONC&EP appears to be around 0.7. How was the conclusion drawn that this is 15%, 27%, and 36% lower than other models? Additionally, could you clarify the relationship with Figure 4 again? ("Specifically, in Figure 4, the throughput for RA-DQL-ONC&EP is depicted as 72% out of 1000 time steps.")

5. It was mentioned that RA-DQL-ONC&EP shows superior performance compared to other models. What about its computational overload and time? How does this impact its application?

Reviewer 2 ·

Basic reporting

Clear and unambiguous, professional English used throughout.
Literature references, sufficient field background/context provided.
Professional article structure, figures, tables. Raw data shared.
Self-contained with relevant results to hypotheses.
Formal results should include clear definitions of all terms and theorems, and detailed proofs.

Experimental design

Research question well defined, relevant & meaningful. It is stated how research fills an identified knowledge gap.
Rigorous investigation performed to a high technical & ethical standard.
Methods described with sufficient detail & information to replicate.

Validity of the findings

Impact and novelty not assessed. Meaningful replication encouraged where rationale & benefit to literature is clearly stated.
All underlying data have been provided; they are robust, statistically sound, & controlled
Conclusions are well stated, linked to original research question & limited to supporting results.

Additional comments

The paper is generally well-written. On the other hand, the paper should be revised based on the following issues:
+ The main contributions of the paper should be given clearly.
+ The organization of the paper should be given clearly.
+ Preamble information should be given between section "4. Deep Q-Network Model" and subsection "4.1 State Representation:".
+ The tables and figures should be given in the text instead of being given at the end of the manuscript.
+ The related work should be improved by adding the more recent paper considering similar problems.

---

## Round 0.2 · accepted · Accept

Dear Authors, congratulations!

Reviewer 1 ·

Basic reporting

My requests have been well reflected.

Experimental design

No comment

Validity of the findings

No comment